# Quantifying inter-annual variability on the space-use of parental Northern Gannets (*Morus bassanus*) in pursuit of different prey types

**Kyle J. N. d'Entremont**[1]*, **Isabeau Pratte**[2], **Carina Gjerdrum**[2], **Sarah N. P. Wong**[2], **William A. Montevecchi**[1]

**1** Cognitive and Behavioural Ecology Program, Psychology Department, Memorial University of Newfoundland, St. John's, Newfoundland and Labrador, Canada, **2** Canadian Wildlife Service, Dartmouth, Nova Scotia, Canada

* kjde08@mun.ca

**Data Availability Statement:** The entire minimal underlying dataset pertinent for this manuscript is publicly available in a data package on the

## Abstract

Spatial planning for marine areas of multi-species conservation concern requires in-depth assessment of the distribution of predators and their prey. Northern Gannets *Morus bassanus* are generalist predators that predate several different forage fishes depending on their availability. In the western North Atlantic, gannets employ different dive tactics while in pursuit of different prey types, performing deep, prolonged U-shaped dives when foraging on capelin (*Mallotus villosus*), and rapid, shallow, V-shaped dives when foraging on larger pelagic fishes. Therefore, much can be inferred about the distribution and abundance of key forage fishes by assessing the foraging behaviour and space-use of gannets. In this study, we aimed to quantify space-use and to determine areas of suitable foraging habitat for gannets in pursuit of different prey types using habitat suitability models and kernel density utilization distributions. We deployed 25 GPS/Time-depth recorder devices on parental Northern Gannets at Cape St. Mary's, Newfoundland, Canada from 2019 to 2021. To assess the influence of environmental variables on gannets foraging for different prey types, we constructed three different habitat suitability models: a U-shaped dive model, and two V-shaped dive models (early and late chick-rearing). Suitable foraging habitat for capelin, deduced by the U-shaped dive model, was defined by coastal, shallow waters with flat relief and sea surface temperatures (SST) of 11–15˚ C. Suitable habitat for early V-shaped dives was defined by shallow and coastal waters with steep slope and SST of 12–15˚C and ~18˚C, likely reflecting the variability in environmental preferences of different prey species captured when performing V-shaped dives. Suitable habitat for late V-shaped dives was defined by shallow coastal waters (<100m depth), as well as waters deeper than 200 m, and by SST greater than 16˚C. We show that space-use by gannets can vary both within and between years depending on environmental conditions and the prey they are searching for, with consequences for the extent of potential interaction with anthropogenic activities. Further, we suggest regions defined as suitable for U-shaped dives are likely to be critical

biotelemetry data repository "MoveBank" at the following DOI: https://doi.org/10.5441/001.1.5km7v2s3.

**Funding:** WAM received grant #2018-06872 from the Natural Sciences and Engineering Research Council of Canada Discovery Grant program. The funder played no role in study design or the publication process. URL: https://www.nserc-crsng.gc.ca/professors-professeurs/grants-subs/dgigp-psigp_eng.asp WAM received the sub-grant # 57177 to Memorial University of Newfoundland and Labrador from the Fisheries and Oceans Canada Coastal Environmental Baseline Program. The funder played no role in study design or the publication process. URL: https://www.dfo-mpo.gc.ca/science/partnerships-partenariats/research-recherche/cebp-pdecr/index-eng.html.

**Competing interests:** The authors have declared that no competing interests exist.

habitat of multi-species conservation concern, as these regions are likely to represent consistent capelin spawning habitat.

## Introduction

Anthropogenic stressors are negatively impacting marine ecosystems worldwide, particularly in coastal environments where human activity is most concentrated [1]. Marine birds are susceptible to harm from many of these stressors, including light pollution [2, 3], chronic oil pollution [4, 5], and depleted forage fish populations due to overfishing and climate-induced shifts in distribution, phenology, and regional predictability [6–10].

Coastal southeastern Newfoundland (NL) is a region where a high degree of anthropogenic activity and large populations of marine birds overlap. In this region, anthropogenic risks of greatest concern for marine birds include marine traffic (fisheries, shipping, oil transport, tourism, and recreational use), bycatch in longlines and gillnets, light pollution, and chronic ship-source oil pollution [2, 11–13]. Climate change and overfishing have also been linked to massive declines in biomass and recruitment success of three key forage fish species in this region: Atlantic mackerel (*Scomber scombrus*, [14, 15]), Atlantic herring (*Clupea harengus*, [16]) and capelin (*Mallotus villosus*, [17]).

The Northern Gannet (*Morus bassanus*), the largest breeding seabird in the North Atlantic, is a generalist, opportunistic feeder, that preys on a wide range of forage fish species depending on availability, which changes dramatically within and between years [18–20]. Northern Gannet colonies in the western North Atlantic, including the southernmost colony in the world at Cape St. Mary's, NL, have exhibited record low productivity in recent years, and plateaued population size, which have been associated with declines in prey availability due to overfishing and climate change [8, 9, 21].

Northern Gannets use different foraging tactics depending on the prey type they are foraging upon. In the western North Atlantic ecosystem, these differences are expressed quantifiably by dive profiles, with prolonged, deep, U-shaped dives associated with pursuit and capture of capelin, while shallow, V-shaped dives are associated with rapid capture of mackerel, Atlantic saury (*Scomberesox saurus*) and Atlantic herring [19, 20, 22, 23]. Northern Gannets breeding at Cape St. Mary's, NL, demonstrate distinct shifts in dive profiles, with U-shaped dives falling out of favor to almost exclusively using V-shaped dives as the breeding season progresses [20].

Habitat suitability models have been used broadly to identify critical habitat for several seabird species (e.g., [24–26]). Modelling habitat suitability using locations of different dive profiles allows us to delineate key foraging areas and how they might vary temporally and with the pursuit of different prey types. We propose that these models can also be used as proxies to identify key habitat areas and environmental drivers for the distribution of these key forage fish.

Our main objectives were to 1) identify key foraging areas for Northern Gannets in southeastern Newfoundland, and 2) determine if Northern Gannet foraging areas differed when using different foraging tactics (*e.g.*, dive profiles). From this assessment, we aimed to provide information on how environmental factors might influence the distribution of Northern Gannets and their prey, for a better understanding of critical areas of multi-species conservation management concern in southeastern Newfoundland.

## Methods

### Study area and species

Cape St. Mary's Ecological Reserve (46.81˚N, 54.18˚W) lies west of Placentia Bay, east of St. Mary's Bay, and north of the southeastern Grand Banks at the southwestern tip of the Avalon Peninsula in eastern Newfoundland, Canada (Fig 1). This site is home to 14,598 breeding pairs of Northern Gannets as of 2018, which represents approximately 14% of the Eastern Canadian population [9, 27]. Since 2010, population size has plateaued, and productivity has been low, like that observed at the largest North American colony located in Quebec (Bonaventure Island, [9]). Average incubation time of gannets in the western North Atlantic is ~44 d (late April to late June) with an average chick-rearing period of ~90 d, ranging from early June to October [28].

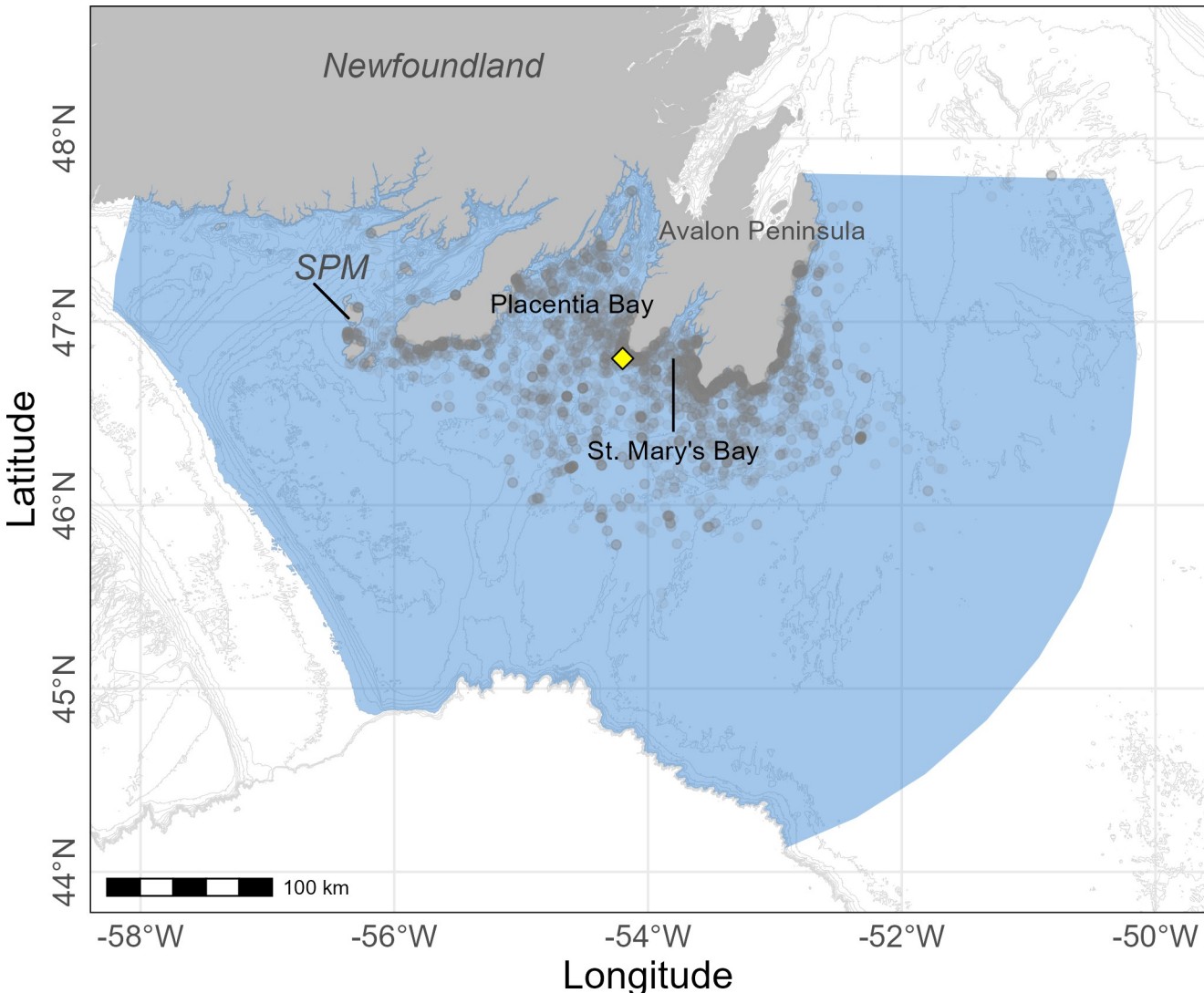

**Fig 1. Foraging domain (blue) used in the habitat modelling for parental Northern Gannets breeding at Cape St. Mary's, Newfoundland, Canada (yellow diamond).** Isobaths (grey lines) are at 50 m intervals. Grey dots are all diving locations recorded in 2019, 2020, and 2021. SPM = Saint-Pierre-et-Miquelon.

## GPS device deployment

Ten gannets in 2019, seven gannets in 2020, and eight gannets in 2021 were fitted with either battery (n = 7) or solar-powered (n = 18) Ecotone Uria 300 GPS loggers with Temperature-Depth Recorders (TDR). Five devices were deployed on both July 18, 2019, and August 21, 2019. Seven devices were deployed on July 18, 2020. Eight devices were deployed on July 15, 2021. Tagged birds had chicks that were ~2–4 weeks old when devices were deployed. Tags weighed 13.5 g (dimensions: 36x22x12.5 mm) and were attached to the four innermost rectrices just below the uropygial gland with Tesa ℝ tape and cable ties. Birds were captured using extendable noose poles, weighed with a 5 kg Pesola ℝ spring scale, and equipped with Canadian Wildlife Service aluminum bands on their right legs. Mean mass of tagged gannets across all three years was 3547 ± 459 g. GPS devices were < 0.5% of body mass for all individuals and the risk of negative tag effects was minimized [29]. Tags were set to record the location of each bird every 15 min. Dive depth was recorded every 1 s after submersion during dives (i.e. from first entry into water until resurfacing). An Ecotone base station with a directional antenna was installed ~50 m away from the colony to upload data from devices when birds returned to the colony. The detection range of the base station varies from 200–500 m, depending on environmental conditions. Devices were set to only record GPS locations and dive information when out of range of the base station to conserve battery life. Gannets were handled under the Canadian Wildlife Service permit 10332K and Memorial University of Newfoundland Animal Care Committee animal care permit 19-01-WM.

## Data processing

All data processing and subsequent analysis were conducted using the statistical software "R" version 4.04 [30]. Dive depth, duration, and bottom time from GPS-tagged gannets were determined for each dive using the package 'diveMove' [31]. Dives that were less than 1 m in depth were removed from further analysis, as these may have been associated with bathing bouts and unlikely to be in pursuit of prey. All dives which had a bottom time ≥ 3 s, and/or total duration > 10 s and a depth > 8 m were designated as "U" dives and all dives which had a bottom time < 3 s, and/or durations < 10 s and depth < 8 m were considered "V" dives (see [21]). Foraging locations were considered to be any GPS location recorded within 30 min before a dive (e.g., [20]). Although U-shaped dives peak during early chick-rearing, V-shaped dives are performed by gannets throughout the chick-rearing period. To account for the differing prey species in the region in early and late chick-rearing and the possibility that early V-shaped dives could be associated with herring rather than mackerel, or singular capture of capelin (rather than a U-shaped dive for multiple fish), we divided V-shaped dives in two categories: early and late V-shaped dives. We used the marked drop in daily proportions of U-dives as the cut-off date to classify early vs late V-shaped dives (Fig 2; 2019–16-Aug, 2020 and 2021–10-Aug).

## Dive distributions

We tested for independence between year and dive shape using a Chi-square test to determine if there was an association in the distribution of dive shapes among years. The spatial distribution of each dive type was analyzed using the package 'track2KBA' [32] to estimate 50% and 95% kernel density utilization distributions (UDs). The 50% UD represents core areas of diving locations (e.g., where 50% of dive locations are concentrated, whereas the 95% UD represents the range of diving activity across the animals' home range). We used the function findScale, which uses First Passage Time analysis to identify the spatial scale of movement at which area-restricted search is occurring within individual trips. This function returns a value corresponding to the log of the average foraging range (mag) and is considered a conservative

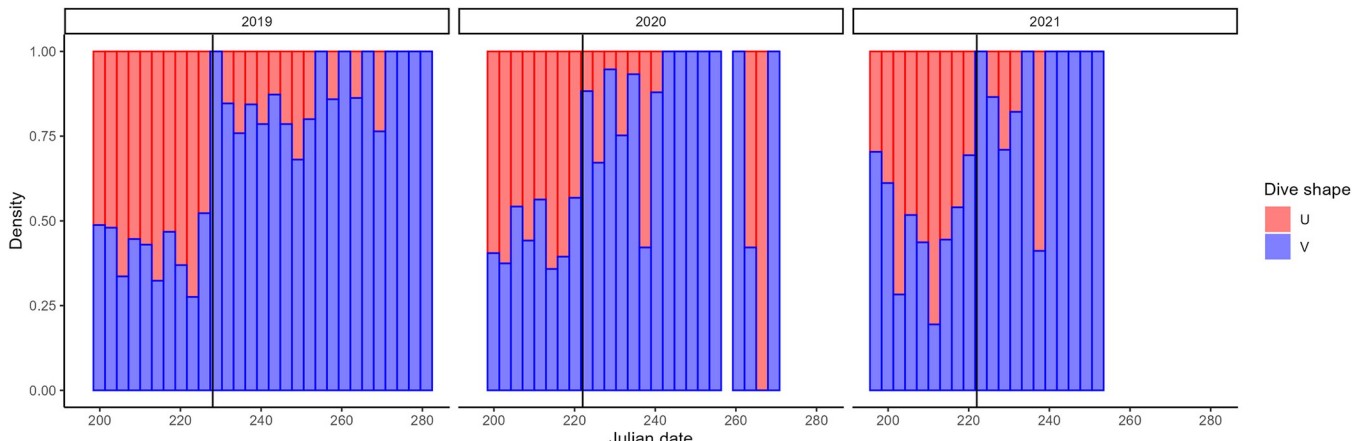

**Fig 2. Daily proportion of U- and V-shaped dives performed by Northern Gannets.** The black vertical line indicates the cut-off point used to classify V-shaped dives occurring early or late (2019–16-Aug, 2020 and 2021–10-Aug).

approach to the estimation of utilization distribution of central place foragers [32]. However, given that we were interested in estimating utilization distributions of diving locations distinctly considered from the animals' full trajectories, the scale at which those occur was not comparable to the scale at which central place foraging occurs in relation to an animals' complete movement along a foraging trip. Thus, we multiplied the log of the average foraging range by 2 for our smoothing term h (8 km) to achieve better coverage when considering the 95% UD and ensure we were not under-representing the area used by foraging gannets. The purpose of using the kernel estimator in our case was to visually represent the distribution of the different dive shapes across years. Since prey distribution and availability are driven by environmental conditions [33–36], we assume the two dive profiles reflect prey type availability [37]. Thus, we built three habitat suitability models for each type of dive (U-, early V-, late V-shaped dives).

## Habitat suitability modelling

For the computation of habitat suitability models, we extracted environmental variables from a domain representative of the foraging area of parental Northern Gannets as determined from GPS tracks. This domain was defined by the farthest eastern and western distances from the colony reached by an individual performing a dive multiplied by 1.1 (451 km), then by drawing an arc until reaching the continental shelf drop in the south, and until reaching the latitude of the northernmost location reached by a gannet (see Fig 1). In each year and for each dive type, three random locations (hereafter "pseudo-absences") were generated within the domain for each GPS device derived foraging location (hereafter "presence") using the *spsample* function in the R package 'sp' ([38], see also [24, 39]). Each pseudo-absence was randomly assigned a date within the range of the study period each year. To build habitat suitability models for each dive type, we included the following covariates: sea surface temperature (SST,˚C resolution of 0.01˚, daily), depth (GEBCO, m; resolution of 0.004˚), slope, distance to colony (km), latitude and longitude. Chlorophyll a was also considered but removed from further models due to poor spatiotemporal coverage over the study period. SST data were retrieved from the Multi-scale Ultra-high Resolution SST Analysis fv04.1 dataset on the Environmental Research Division's Data Access Program (ERDDAP, https://coastwatch.pfeg.noaa.gov/erddap/griddap/jplMURSST41.html). For each presence and pseudo-absence, values for all

environmental variables were extracted using the package 'raster' [40], and distance from the colony (km) was determined.

Given that autocorrelation is inherent to tracking data and that individuals may have different habitat preferences, fitting random slopes for individuals might be useful. However, differing random effect levels prevent mixed effects models from predicting into new datasets [41, 42]. Thus, in our case, GAMs were a practical modelling approach and individuals were not considered as random effects due to the impracticability given our predictions. For each dive type, we determined which variables to include in our final model by constructing a generalized additive model (GAM) with each variable included as the smooth parameters of the model and using penalized thin-plate regression spline. Latitude and longitude were smoothed using a two-dimensional spline. Year was also added as a fixed term in the model. The binomial response was true location (presence, 1), and the randomized background points (pseudo-absence, 0). We first tested for pairwise concurvity in the model components using the concurvity function of the 'mgcv' package (version 1.8–42; [43]). Then, for each dive type, automated model selection was performed on the global model using the dredge function of the package "MuMIn" (version 1.45.7; [44]). Models were compared using Akaike information criterion (AICc) and we selected for our final predictive model the one with the smallest AICc (Table 2).

For the duration of the three-year tracking period, we predicted the daily probability of occurrence of each dive shape within the domain defined for Northern Gannets breeding at Cape St. Mary's. The mean number of dives per day across all three years was 70.8 ± 4.7 SE (n = 199), though the number of dives per day consistently decreased as the breeding season progressed in each year (Fig 3). A grid was overlaid over the domain extent at a resolution corresponding to the lowest common resolution of our final environmental layers (0.01˚, ~1000 m for the SST layer). For each day of the study period, we extracted values from the covariates included in our final models at the centroid of each grid square. We predicted the probability

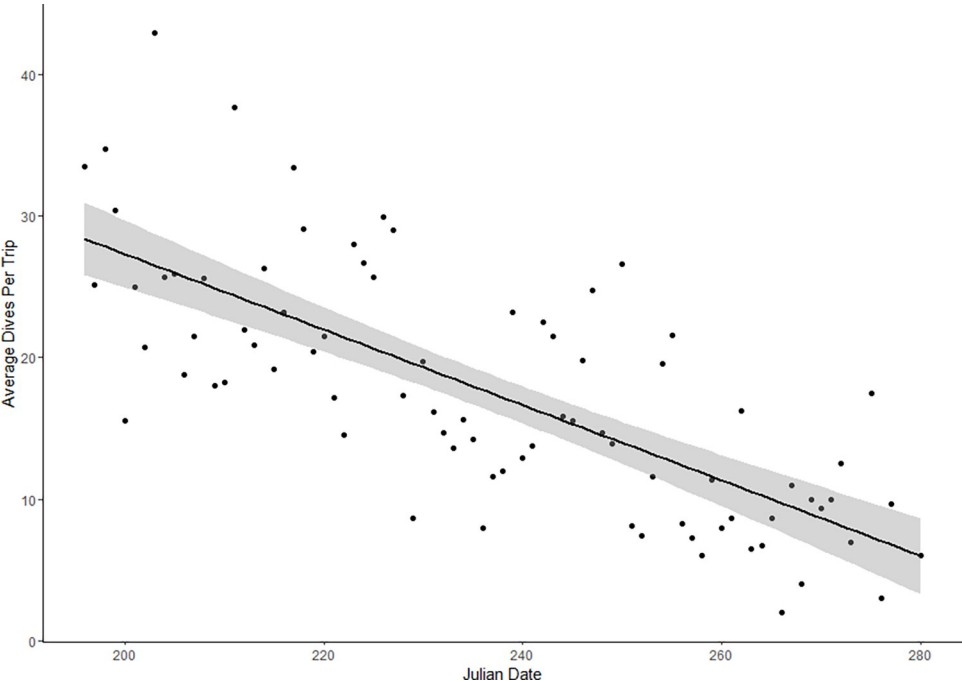

**Fig 3. Mean number of dives per day (Julian Date) performed by all individual gannets across all three years.**

of occurrence for U, early and late V-shaped dives using the function *predict.gam* ('mgcv' package) for each day of the tracking study (predicted within the respective time spans for early and late V-dives), obtaining a daily habitat suitability index (HSI) for each dive type. We averaged the resulting HSI raster layers to generate a final predictive map reflecting the average daily probability of occurrence of U, early and late V-shaped dives, and the error of the spatial prediction was summarized by calculating the pixel-specific standard deviation. Given that the only dynamic variable used was SST, minimal differences were noted when using the model to predict HSI for each of the three study years, thus we presented an overall model prediction by averaging daily predictions obtained over the three years.

## Results

### Dive distributions

There was significant temporal variation in the distributions of U- and V-shaped dives among years ($\chi^2$ = 122.78, $p < 0.001$). For example, fewer U-shaped dives were performed by gannets than expected in 2019 and 2021 (Table 1). There were fewer than expected V-shaped dives during early chick-rearing and more than expected during late chick-rearing in 2019, but this effect was reversed in 2021. Only in 2020 were more U-shaped dives recorded than expected, and generally fewer V-shaped dives were observed that year (Table 1).

The spatial distribution of U-shaped dives by Northern Gannets appeared consistent across years, with 50% UD areas centered around the colony but extending slightly further east in 2021 (Fig 4). Early V-shaped dives had an extended distribution compared to the area used for U-shaped dives, especially in 2021 (Fig 4). The extent of late V-shaped dives was much wider-ranging than the other two dive types, especially in 2019 and 2020 when foraging gannets travelled west to Saint-Pierre-et-Miquelon and to Fortune Bay.

### Model predictions

Response to the suite of environmental variables included in our models predicted that suitable habitat for late V-shaped dives was more extensive and diffuse than for U-shaped and early V-shaped dives (Fig 5), for which the probability of occurrence was more clustered around the southern coast of the Avalon Peninsula. Although early V-shaped dives co-occurred with U-shaped dives, the former were more likely to be farther from the colony, over Placentia Bay, and generally in offshore areas south of the Avalon Peninsula (Fig 5B). U-shaped dives were strikingly more coastal (Fig 5A). Overall, the predictive habitat models for each type of dive captured the variability observed in the foraging distribution of Northern Gannets across years, but also highlighted areas of lower density (outside the 50% UD).

After checking for concurvity of our global model components, we discarded the covariate distance to the colony from the model for each dive type as concurvity with the other covariates included as smooth terms (latitude/longitude, slope, SST) was high ($> 0.9$). Following

**Table 1. Sample size and tracking period (when dives were recorded) for each year during which Northern Gannets were tracked from Cape St. Mary's, NL.** Number of observed dives of each shape in each year, and number of expected dives of each shape in each year and their residuals following Chi-square test ($\chi^2$ = 122.78, $df$ = 4, $p < 0.001$).

| Year | N ind. | Tracking period | Observed | | | Expected | | | Residuals | | |
|---|---|---|---|---|---|---|---|---|---|---|---|
| | | | U | V-early | V-late | U | V-early | V-late | U | V-early | V-late |
| 2019 | 10 | 19-Jul to 29-Sept | 347 | 1074 | 748 | 419 | 1200 | 551 | -3.5 | -3.6 | 8.4 |
| 2020 | 7 | 18-Jul to 26-Sept | 543 | 1075 | 414 | 392 | 1124 | 516 | 7.6 | -1.5 | -4.5 |
| 2021 | 8 | 15-Jul to 09-Sept | 138 | 797 | 190 | 217 | 622 | 286 | -5.4 | 7 | -5.7 |

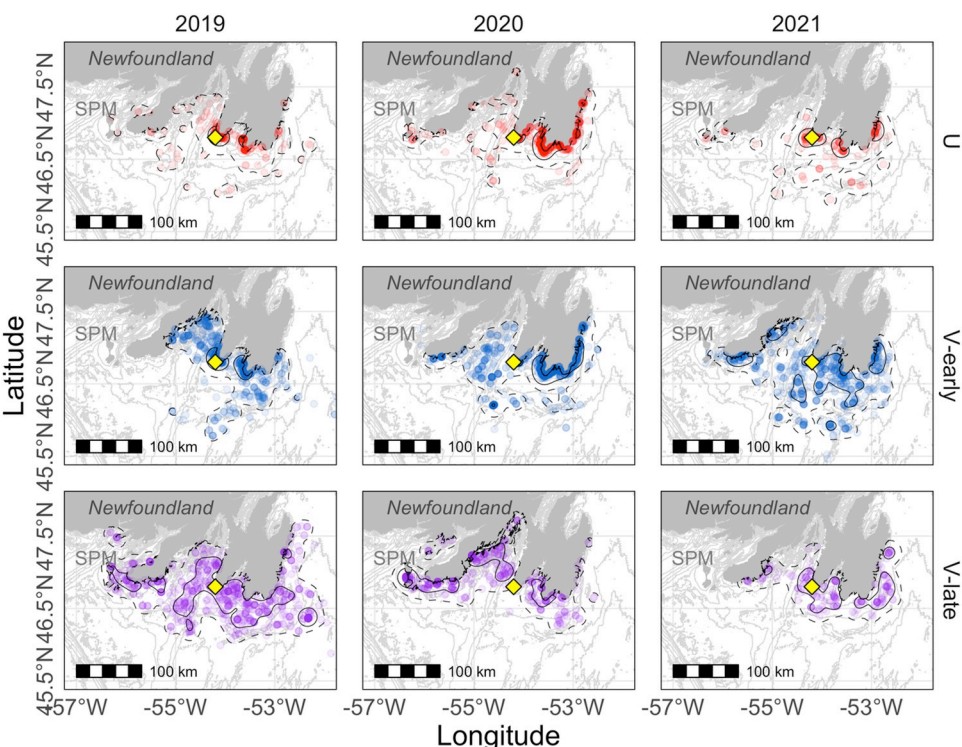

**Fig 4. Spatial distribution of U- (red), early (blue) and late (purple) V-shaped dives performed by Northern Gannet tracked from Cape St. Mary's (yellow diamond) for each year of the study.** Utilization distributions are shown by the solid line (50% UD) and the dashed line (95% UD), smoothing parameter h = 8 km. Saint-Pierre-et-Miquelon (SPM).

model selection, the final GAMs retained all the covariates tested (bathymetry, latitude/longitude, slope, and SST) except the model for late V-dives, which did not include slope (Table 2). For U-shaped dives, the model explained 76.7% (adjusted $R^2$ = 0.78) of the total deviance. The model for early V-shaped dives explained 67.8% (adjusted $R^2$ = 0.68) of the total deviance, while the model for late V-shaped dives explained 60.2% (adjusted $R^2$ = 0.61) of the total deviance.

The partial response curves from the GAM models indicated that the probability of occurrence of U-shaped dives decreased between sea surface temperatures spanning from about 14°C to 18°C (Fig 6A), while probability of occurrence of V-shaped dives during both early and late chick-rearing increased at temperature beyond 16°C and this was slightly more pronounced for V-shaped dives occurring later (Fig 6B and 6C). V-shaped dives were more likely to occur over a broader area compared to U-shaped dives which were more constrained in space and more likely to occur around 54°W and 52°N (Fig 6). Probability of occurrence of U-shaped dives increased slightly when foraging at shallower depths (ca > 75 m). This response was different for V-shaped dives, for which probability of occurrence in relation to depth was positive both for deeper (ca < 200 m) and shallower waters (ca > 100 m). The response to slope was marginal, with fewer data points falling over slopes greater than 8°, but different among dive types and was not a covariate retained for late V-shaped dives. Overall, U-shaped dive occurrence positively responded to flatter sea bottom (0–5°slope, Fig 6A), although this response was marginal. Early V-shaped dives had higher positive response to steeper slope, but also had greater error owing to fewer data points falling over steeper slope.

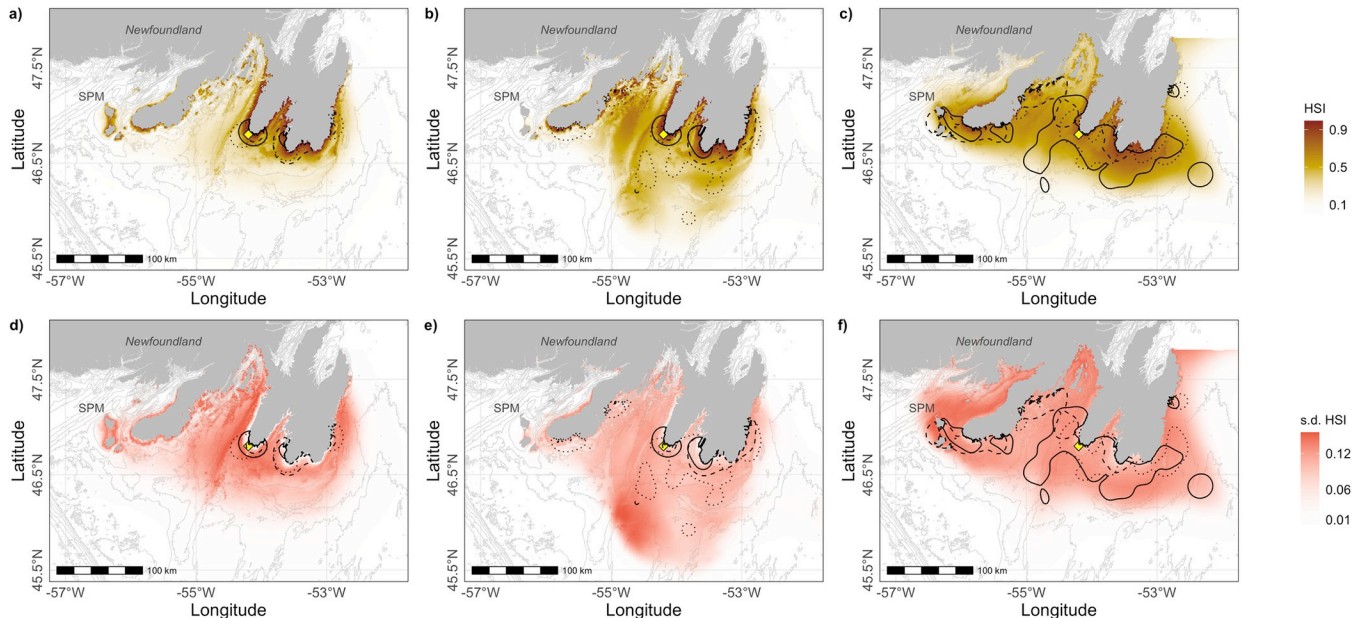

**Fig 5.** Spatial habitat predictions for a) U-shaped dives, b) early and c) late V-shaped dives of Northern Gannets tracked from Cape St. Mary's (yellow diamond) during three breeding seasons. Average of daily habitat suitability index (HSI) scaled from 0 (unsuitable) to 1 (highly suitable) (a, b, c), and standard deviation of daily HSI for d) U-shaped dives, e) early and f) late V-shaped dives. The spatial resolution of the prediction was set to the lowest resolution of our environmental layers (SST, at 1000m). Utilization distributions at 50% for each dive type are overlaid on the top panels for each year represented by different line types (solid—2019, dashed—2020, dotted—2021). Saint-Pierre-et-Miquelon (SPM).

## Discussion

This study is the first to quantify associations between the foraging of the Northern Gannet on different prey types (*e.g.*, U-shaped dives for capelin; V-shaped dives for mackerel, saury, etc.) and environmental variables using habitat suitability models in the western North Atlantic.

Increased foraging effort by seabirds, including gannets, is often indicative of decreased prey availability [45–48], suggesting increased foraging effort on capelin was required in 2020, perhaps because other prey (i.e., mackerel, saury) were not yet available, or because capelin availability was low. The notion that an increase in U-shaped dives was associated with

**Table 2. Top three candidate models following automated model selection.** Plus signs (+) represent the inclusion of covariate use as smooth terms in the global GAM models. Year was considered as a fixed term in the GAM models. Akaike model weights (*w*) are also shown and further informed model selection. Final models were selected based on the lowest AICc and are represented in bold.

| U-shape | Year | Bathy | Lat:Lon | Slope | SST | df | AICc | delta | *w* |
|---|---|---|---|---|---|---|---|---|---|
| | + | + | + | + | + | **34** | **987.88** | **0** | **0.68** |
| | + | + | + | | + | 31 | 989.61 | 1.73 | 0.29 |
| | | + | + | + | + | 32 | 994.95 | 7.07 | 0.68 |
| **V-shape early** | **Year** | **Bathy** | **Lat:Lon** | **Slope** | **SST** | **df** | **AICc** | **delta** | *w* |
| | + | + | + | + | + | **53** | **4019.12** | **0** | **1** |
| | + | + | + | | + | 45 | 4091.82 | 72.7 | 0 |
| | | + | + | + | + | 51 | 4105.98 | 86.86 | 0 |
| **V-shape late** | **Year** | **Bathy** | **Lat:Lon** | **Slope** | **SST** | **df** | **AICc** | **delta** | *w* |
| | + | + | + | | + | **39** | **2450.27** | **0** | **0.72** |
| | + | + | + | + | + | 34 | 2452.18 | 1.9 | 0.*28* |
| | + | | + | | + | 37 | 2482.87 | 32.59 | 0 |

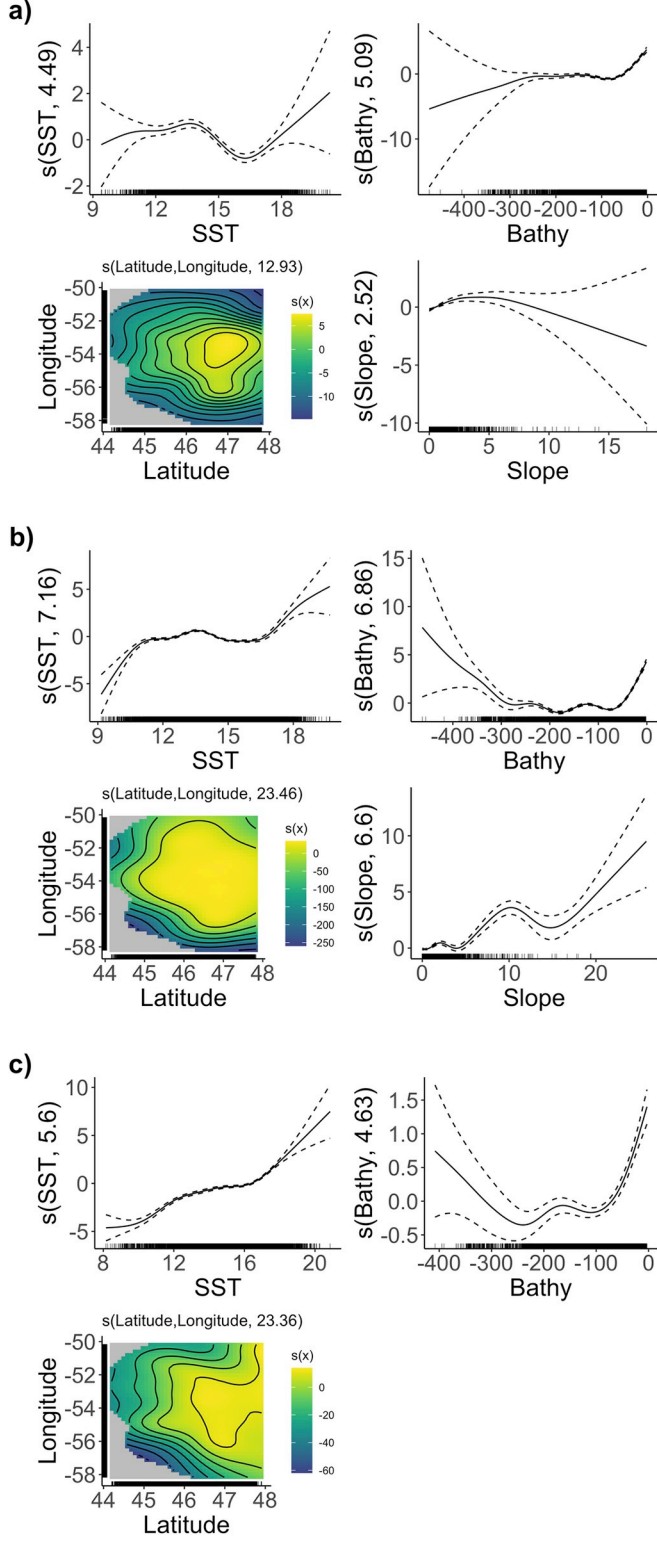

**Fig 6.** Model variables for estimated probability of occurrence of a) U-shaped dives and b) early and c) late V-shaped dives by Northern Gannets tracked during three breeding seasons (2019–2021) at the colony of Cape St. Mary's, Newfoundland. The y-axis represents the function of each predictor variable with the effective degrees of freedom (edf) of the smooth term in brackets. 0 on the y-axis corresponds to absence of an effect of the predictor variable on the estimated probability of occurrence. Environmental variables included as smooth terms in the GAMs were daily sea

surface temperature (SST; ˚C), bathymetry (Bathy; m), slope (˚), and longitude and latitude (˚) considered as 2-dimensional spline in the smooth function; the y-axis scale varies to emphasise model fit. The confidence intervals (dashed lines) are shown at 1 standard error above and below the smoothed estimate.

lowered prey availability is consistent with the fact that capelin availability (e.g., biomass and spawning locations) in Newfoundland waters was lower than usual in 2020 [20, 49]. The number of V-shaped dives was greater than expected in both 2019 and 2021, suggesting that there may have been increased effort associated with lower availability of mackerel and saury in these years [9]. As capelin is a key prey species for gannets during early chick-rearing due to the size and energetic requirements of chicks [50], this might explain why U-dives were greater than expected when capelin availability was low in 2020. The reliance on capelin in early chick-rearing due to its energy density and size of the chick (e.g., prey larger than the smaller capelin may not be consumable by the smaller chicks at this stage) constrains foraging plasticity at this critical time.

## Habitat suitability in pursuit of varying prey types

The U-shaped dive model suggested that suitable habitat for capelin was constrained towards coastal shallow waters with flat bottom relief, with SST ranging between ~11–15˚C. Plus, the probability of occurrence decreased as SST increased above 15˚C, which is consistent with the tendency for capelin to only spawn in waters below 12˚C [51, 52]. Moreover, the decreasing probability of occurrence of U-shaped dives as seafloor slope increases is consistent with the typical spawning site characteristics of capelin in coastal Newfoundland, i.e., flat bottom relief and/or depressions on the seafloor confined by gently sloping marine trenches [53]. This further strengthens previous assertions that U-shaped dives by Northern Gannets in the western North Atlantic are primarily associated with foraging on spawning capelin [20, 22, 23]]. Furthermore, areas deemed highly suitable for capelin foraging by gannets overlaps with putative beach and deep-water spawning areas identified by capelin fishers with extensive knowledge and experience of the local distribution of this species [54]. Thus, as U-shaped dives tend to be specific to capelin in this ecosystem, these habitat suitability models could be considered as a proxy for suitable habitat for capelin. More importantly, the areas deemed highly suitable for capelin within this study are also likely to be critical regions of multi-species management concern as many aquatic (e.g. Atlantic cod *Gadus morhua*) and avian predators (e.g., black-legged kittiwakes *Rissa tridactyla*, common murres *Uria aalge*, Atlantic puffins *Fratercula arctica*) heavily rely on capelin [55–57].

The early V-shaped dive model suggested that during early chick-rearing, gannets are foraging in areas with shallow waters, over steeper slopes, and constrained to SST ranging from 11–15˚C or above 18˚C. Thus, the effect of SST on early V-shaped dive occurrence is possibly linked with the water temperature preferences of prey species other than capelin; indeed, mackerel prefer temperatures of 9–13˚C while saury tend to inhabit warmer waters of ~18˚C [58, 59]. Therefore, it is plausible that early V-shaped dives are used when foraging for mackerel and saury as they start to migrate into the region in late July [60, 61]. As well, this SST range also includes the thermal preference for adult herring (8–12˚C, [62]), meaning that V-shaped dives could also indicate foraging on herring. Some V-shaped dives were associated with steeper slopes; and it is unlikely that gannets were foraging for spawning capelin at those sites given capelin prefer areas with flat bottom relief [51, 52]. However, it is possible that some V-shaped dives were in pursuit of small aggregations of capelin and/or capelin migrating to deeper waters away from their spawning grounds [63] given that V-shaped dives also had a slightly higher chance of occurring over flatter bottom relief. Overall, both early and late V-

shaped dive models revealed that probabilities of occurrence of both dive types were higher over a broader geographic area, which was reflected in the utilization distribution of both dive types compared to U-shaped dives. Key habitat during late chick-rearing covered a broader geographic range within the study region, that was characterized by relatively shallow waters, but also to a higher probability of utilizing deeper waters, likely as a result of range expansion offshore. Later in the season, as SST increases and capelin abundance decreases, gannets forage on other prey [20], such as saury and mackerel which prefer warmer waters [58, 59].

Overall, the predictions obtained from the models built for each dive type captured the variability observed in the foraging distribution of Northern Gannets across years, and highlighted areas of lower density (outside the 50% UD). For example, this might suggest that the west and inner coast of Placentia Bay represents more favorable habitat for gannets performing U-shaped and early V-shaped dives. Considering the inter-annual variability in space use, such an area might be used in some years, and not others. Finer scale environmental predictors and better knowledge of prey distribution and behaviour would be beneficial, especially for V-shaped dives during late chick-rearing, for which the model explained only 60.2% of the total deviance. V-shaped dives are performed for more than one prey type that could have different habitat preferences, thus making it difficult to capture in a single model. Nonetheless, the results from our models built with different dive types are mirroring what we know about the temperature tolerances of key forage fishes, and gannet behavioural responses to those [7, 64].

## Risk associated with anthropogenic activity and climate change

Space-use by gannets ranging from Cape St. Mary's depends on exploitation of different prey bases and their breeding and migratory phenology, so the anthropogenic pressures gannets face in southeastern Newfoundland can differ both intra- and inter-annually. During early chick-rearing (July/early August), gannets typically foraged in coastal waters, while during late chick-rearing, adults were less constrained to the coast and extended their foraging distribution towards more pelagic waters. These patterns would place them at greater risk of encountering gillnets associated with inshore cod and Atlantic herring fisheries [12] during early chick-rearing, and more likely to interact with large vessels (e.g., oil tankers) passing through shipping lanes during late chick-rearing [13, 65].

Given projections of warming ocean climate [66], marine ecosystems such as the western North Atlantic will experience further shifts in the phenology and distribution of critical prey species [67–69], and subsequently, behaviour, foraging distribution, and breeding success of marine top predators [70]. Hence, it is expected that gannets will have to expend greater energy on foraging to compensate for changes in availability, as has already been observed in the western North Atlantic during years of poor capelin availability ([20]; see also [46]).

Capelin has exhibited a 30-fold decline in biomass and delayed spawning in the western North Atlantic since the early 1990s [17, 35]. Continued declines in capelin biomass could occur with earlier sea ice retreat that results in earlier spring bloom of primary producers, ultimately impacting copepods—the key prey of capelin [35]. Such mismatches could represent a cascading effect through the food web creating shortages and mismatches of capelin availability with the food requirements of parental seabirds provisioning offspring [17, 35]. Ocean temperature is the most predictive environmental variable of the horizontal and vertical distribution of capelin from June to November within the study region (e.g., warmer waters linked to decreased presence; [71], with the importance of ocean temperature on capelin distribution being greatest in August. This influence of warming waters on capelin distribution is critical as it completely overlaps with the chick-rearing period of Northern Gannets breeding at Cape St. Mary's. If current warming trends continue, capelin distributions could shift

northwards and to deeper waters during the gannets breeding season, reducing prey availability. However, as gannets exhibit foraging plasticity within and between years (e.g., [20, 64, 72]), they may be able to compensate for these distributional shifts by exploiting other prey species. Along with continued demographic monitoring at the colony to measure population-level impacts, further monitoring of foraging behaviour of parental gannets at Cape St. Mary's is warranted to assess ocean climate responses to these distributional shifts.

The spawning distribution of Atlantic mackerel is also expected to shift northwards in the coming decades [67], with a significant overlap likely to occur within the foraging range of Cape St. Mary's gannets during incubation (May-June) and early (July) chick-rearing period [20]. This shift could cause temporal and spatial mismatches between gannets and mackerel, with mackerel becoming available during early chick-rearing, and limited in late chick-rearing as sea water temperature rise above the species' thermal threshold of ~15–16°C [7, 9, 59]. As mackerel is a critical prey species for Northern Gannets during late chick-rearing [50], such a shift could have consequences for gannet reproductive success if they are unable to supplement their diet with alternative prey sources. Such supplementation of alternate prey sources could be achieved by a potential increase in saury availability if average sea surface temperatures of ~18°C (the preferred temperature of Atlantic saury, [58]) are reached during the late chick-rearing period. It remains to be seen how gannets might respond to such drastic ecosystem shifts. Warming ocean temperatures in spring/early summer could also cause poor herring spawning conditions and induce a shift from spring spawning to mainly autumnal spawning in eastern Newfoundland [73]. Such a seasonal shift would likely change herring abundance throughout the chick-rearing period, from being an important prey during early chick-rearing towards being available during late chick-rearing, which would alter the space-use of foraging gannets throughout chick-rearing.

## Conclusions

Northern Gannets are flexible and opportunistic generalist top predators [64] that use different foraging tactics for different prey species and behavioural shifts from exploiting cold-water species in early chick-rearing, to primarily warm-water species in late chick-rearing (e.g., [20]). Knowing these biological patterns, we have shown that space-use by parental Northern Gannets in the western North Atlantic differs depending on the prey species they are foraging on (e.g., inshore, shallow waters for capelin and deeper, more pelagic waters for mackerel and saury, etc.). Further, we demonstrate that examining habitat suitability for organisms employing different foraging tactics is a valuable tool for determining how space-use can change when they are reliant on temporally constrained prey sources.

We also show that Northern Gannets may be susceptible to the negative effects of our warming climate, with foraging habitat suitability being heavily reliant on SST thresholds of their poikilothermic prey. With capelin and herring likely becoming less abundant in early chick-rearing and mackerel likely shifting to be more abundant during early chick-rearing, it is likely that gannets will shift their space-use and prey preferences in Newfoundland as climate change forces distributional and phenological shifts of their prey. These shifts in prey distributions could also change the temporal overlap between gannets and anthropogenic activities (e.g., oil tankers and fishing vessels) and warrants further monitoring in the years ahead.

## Acknowledgments

We must thank several people for their help with this study. The Cape St. Mary's Ecological Reserve staff were invaluable, providing accommodations and logistical support over the years.

We thank staff member Chris Mooney in particular, for his assistance with tagging gannets in the field. We thank Rob Ronconi and Sabina Wilhelm of the Canadian Wildlife Service for providing funding support for purchase of tags. We also thank the Newfoundland and Labrador Parks and Natural Areas Division for allowing us to conduct our research at the Cape St. Mary's Ecological Reserve. Gannets were handled under the Canadian Wildlife Service permit 10332K and Memorial University of Newfoundland and Labrador animal care permit 19-01-WM.

## Author Contributions

**Conceptualization:** Kyle J. N. d'Entremont, Isabeau Pratte, Carina Gjerdrum, Sarah N. P. Wong, William A. Montevecchi.

**Formal analysis:** Kyle J. N. d'Entremont, Isabeau Pratte.

**Funding acquisition:** William A. Montevecchi.

**Investigation:** Kyle J. N. d'Entremont, Isabeau Pratte, Carina Gjerdrum.

**Methodology:** Kyle J. N. d'Entremont, Isabeau Pratte, Sarah N. P. Wong.

**Resources:** William A. Montevecchi.

**Supervision:** William A. Montevecchi.

**Visualization:** Kyle J. N. d'Entremont, Isabeau Pratte, Sarah N. P. Wong.

**Writing – original draft:** Kyle J. N. d'Entremont.

**Writing – review & editing:** Kyle J. N. d'Entremont, Isabeau Pratte, Carina Gjerdrum, Sarah N. P. Wong, William A. Montevecchi.

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
