## [Decision Letter · Decision Letter 0]

28 Feb 2023

PONE-D-23-00624Quantifying space-use of parental Northern Gannets (Morus bassanus) in pursuit of different prey typesPLOS ONE

Dear Dr. d'Entremont,

Thank you for submitting your manuscript to PLOS ONE. After careful consideration, we feel that it has merit but does not fully meet PLOS ONE’s publication criteria as it currently stands. Therefore, we invite you to submit a revised version of the manuscript that addresses the points raised during the review process.

We look forward to receiving your revised manuscript.

Kind regards,

Vitor Hugo Rodrigues Paiva, Ph.D.

Academic Editor

PLOS ONE

Journal Requirements:

   "We must thank several people for their help with this study. The Cape St. Mary’s Ecological Reserve staff were invaluable, providing accommodations and logistical support over the years. We thank staff member Chris Mooney in particular, for his assistance with tagging gannets in the field. We also thank the Newfoundland and Labrador Parks and Natural Areas Division for allowing us to conduct our research at the Cape St. Mary’s Ecological Reserve. This study was 

funded by the Natural Sciences and Engineering Research Council and an Ocean and Freshwater Contribution Program grant from Fisheries and Oceans Canada to WAM. Additional funding was provided by NSERC (WAM) and Memorial University of Newfoundland and Labrador. Gannets were handled under the Canadian Wildlife Service permit 10332K and Memorial University of Newfoundland and Labrador animal care permit 19-01-WM."

  "WAM received grant #2018-06872 from the Natural Sciences and Engineering Research Council of Canada Discovery Grant program. The funder played no role in study design or the publication process. URL: https://www.nserc-crsng.gc.ca/professors-professeurs/grants-subs/dgigp-psigp_eng.asp

WAM received the sub-grant # 57177 to Memorial University of Newfoundland and Labrador from the Fisheries and Oceans Canada Coastal Environmental Baseline Program. The funder played no role in study design or the publication process. URL: " ext-link-type="uri" xlink:type="simple">https://www.dfo-mpo.gc.ca/science/partnerships-partenariats/research-recherche/cebp-pdecr/index-eng.html"

Reviewers' comments:

Reviewer's Responses to Questions

**Comments to the Author**

1. Is the manuscript technically sound, and do the data support the conclusions?

Reviewer #1: Partly

Reviewer #2: Yes

Reviewer #3: Yes

Reviewer #4: Yes

2. Has the statistical analysis been performed appropriately and rigorously? 

Reviewer #1: N/A

Reviewer #2: No

Reviewer #3: Yes

Reviewer #4: Yes

3. Have the authors made all data underlying the findings in their manuscript fully available?

Reviewer #1: No

Reviewer #2: Yes

Reviewer #3: Yes

Reviewer #4: No

4. Is the manuscript presented in an intelligible fashion and written in standard English?

Reviewer #1: Yes

Reviewer #2: No

Reviewer #3: Yes

Reviewer #4: Yes

5. Review Comments to the Author

Reviewer #1: Authors,

I enjoyed reading this study about how habitat use models differ based on which behaviour (in this case dive type) is used as a response variable. I think the premise is highly worthwhile, and using the behaviour of gannets to infer the availability of different forage fishes based on foraging mode is really interesting. However, I have some concerns about the methodology and surrounding narrative that I believe need to be addressed. See below for specific comments.

L44-46: This sentence is too complex, maybe split into two, with the latter explaining the depleted forage fish stocks.

L62: Maybe “slowing” instead of “asymptotic” might read a little easier.

L101-104: This section should be represented as a table of deployments.

L120-122: This is a bit confusing: What happens if a bird dives for 10s and has a bottom time of 3s? Could just go by Cox et al. (2016) and solely use bottom time thresholds to separate dive types?

L142-143: I agree that the two types of dive profiles may reflect availability of different prey types, however, this then raises the question of why V-shaped dives were split into early and late? U-shaped dives also occur at almost all times? You mention earlier that U-shaped dives may influence V-shaped dives, but I’m not sure this is really logical. You might need to stick to two different responses (which I would personally opt for), early and late for both dive types, or find stronger justification for the three current responses.

L152-155: I don’t agree with the use of pseudo-absences, when perfectly good absences exist. Why not use locations with no associated dives as the absence in your response? Because dive behaviour is your response, the presence or absence of that behaviour along the track should form your response variable, rather than randomly generated pseudo-absences.

L157: What’s the temporal resolution of SST? Is it MODIS data, and if so, is it level 3 or level 4 type data?

L158: Why not latitude? It may be colinear with SST, but probably worth considering in any case. It would be good to include a 2D lat/lon spline.

L167: Whole model selection using MuMIn::dredge or similar would be much better if feasible.

L170-171: The mgcv::concurvity function is a neat way of looking for multicollinearity within your model. This way is okay too, but the other would be better.

L180-181: If V-shaped dives are being split into early and late, surely they should only be predicted over the time span they are associated with? Please clarify if this is the case or rectify if not.

L188: Maybe association is incorrect here? “Temporal variation in dive types” instead?

L202-204: This spatial variation between early and late V-shaped dives could potentially be good justification for splitting into two responses. However, not sure the pattern is strong enough.

Table 2: Models with nothing but “distance to colony” returning an AUC of 0.9 is a case in point for why pseudo-absences are a poor substitute for genuine absences when trying to determine environmental drivers of behaviour, and not solely distribution. This is evident when adding other environmental variables has a negligible impact on model AUC.

L242-257 figure 5: Model covariates appear a little bit overfit in some cases. In general, you should discuss the model specification in a bit more detail. Are you using cubic or thin-plate splines? Is any shrinkage applied? Shrinkage would be a tidy way to reduce instances of overfitting and return the simplest effective spline. Perhaps transforming heavily skewed variables, such as slope and bathymetry, could be beneficial too, as this might reduce the huge uncertainty around extreme values.

Figure 5: All the bathymetry plots have positive values (assumably on land) with a very high likelihood of dives occurring. Might need to look at how the data are prepared, figure out why this happening, and fix it.

Figure 5: I would also highly recommend the mgcViz package for plotting model outputs from mgcv.

L275-283: Really interesting links here between dive type and prior expectation of the availability of prey. Seems that if a prey type is lacking, they increase the associate dive effort rather than concentrating on other dive types/prey species. Could potentially expand into temporal changes in dietary requirements and why this might prevent plasticity, or might be too complex a topic to just touch on. Just a thought.

L306-308, L321, L370-371: These sentences contradict each other about the temperature preferences of mackerel, or at least seem to?

L346-350: It could be good here to foreshadow the extra energy that gannets may have to expend to target desired prey species? You have a basis for this, given the additional effort in U-shaped dives in the year when capelin was less abundant, and increases in V-shaped dive effort in years with fewer saury and mackerel around. Otherwise, this paragraph seems a bit light and vague for quite an important and relevant point.

L381-387: The conclusion as a whole does not reflect the research that has gone into this paper. Please rewrite, with a focus on the actual findings instead of projecting how space-use will change, which you haven’t really shown.

I hope you find these comments useful.

All the best

Reviewer #2: First of all, I want to congratulate the authors for the great work performed to study the habitat use and spatial distribution of the northern gannet (Morus bassanus) by taking into account their two major diving tactics. Overall, the authors put a considerable amount of effort for habitat suitability modeling exercises, quite complex analyses, and I commend them for this.

Nevertheless, I have a major concern about the modeling approach and some other minor/medium questions that should be taken into consideration before the final approval.

I have added my comments in the word version using track changes, pointing some typographical or grammatical errors, but I provide here the main issues:

INTRODUCTION:

L117: I feel that in this paragraph (L117-L128) should be included more information about the use of habitat suitability models in seabirds. Perhaps authors could indicate some studies where they were successfully applied, even in gannets or other Sulidae species.

L130: Perhaps the ideas in this sentence are in the reverse order. The dive profiles could change when the foraging areas are different, due to the foraging habitat conditions (depth, slope, etc) and to the prey inhabiting these different habitats.

METHODS:

L215: I am concerned about the computation for the estimation of the smoothing parameter (h). Href is a low-conservative method, so possibly we can be dealing with a considerable error that will affect kernel density estimation. I recommend the use of more recent methodologies, such as those in Lascelles et al., 2016 that provide a h value more adapted to your own data.

Lascelles BG, Taylor PR, Miller MGR, Dias MP, Oppel S, Torres L, Hedd A, Le Corre M, Phillips RA, Shaffer SA, Weimerskirch H, Small C (2016) Applying global criteria to tracking data to define important areas for marine conservation. Divers Distrib 22:422–431. doi: 10.1111/ddi.12411

L241-267: Once the individuals performed several dives, I recommend the use of individual as a random factor to avoid pseudo-replication issues. I mean, perhaps an additive mixed model might be more adequate to analyze the data.

Additionally, there are some other gaps that should be clarified in this section:

- Please refer how the pseudo-absences were generated prior modeling exercises.

- It seems that the “year” was not included in the modeling exercises. Since part of this manuscript was focused on the inter-annual variability on the space-use, the “year” must be included as a covariate.

- Have you used smoothed terms in GAM? Perhaps by smoothing SST may be a good way to deal with possible non-linear. distribution of the variable.

- Finally, if smooth terms were used please refer what was the number of thin plate regression splines.

- Besides correlation among the covariates, test for multicollinearity should be evaluated (for instance using the variation inflation factor, VIF), and concurvity if smoothed terms were used.

Two good guides for analyzing ecological data:

Zuur AF, Ieno EN, Smith GM (2007) Analysing ecological data. Springer, New York

Morlini I (2006) On multicollinearity and concurvity in some nonlinear multivariate models. Stat Methods Appl 15:3–26. https://doi.org/10. 1007/s10260-006-0005-9

- Have you used AIC or the ΔAICc? The more correct is using the ΔAICc (the difference in AICc between a given model and the model with the smallest AICc). Nevertheless, I suggest the use of AIC corrected for small sample sizes (AICc), because you were working with a low number of individuals.

Burnham KP, Anderson DR (2002) Model selection and multimodel inference: a practical information-theoretic approach. Springer- Verlag, New York

Additive models are quite useful when working with non-linear variables, such as environmental predictors, however, some assumptions must be fulfilled before model runs.

A good book/guide for the mixed models:

Zuur AF, Ieno EN,WalkerN, Saveliev A a., SmithGM (2009) Mixed effects models and extensions in ecology with R. Springer, New York

RESULTS:

L367: In fact, the threshold is not that clear for me, according to Figure 5a. I would rather say that the U-shaped dives probability of occurrence decreased beyond 150 km from the colony, rather than 200 km.

L368: On the other hand, the early V-shaped dives seemed to decrease beyond 200 km rather than 150 km, while the late V-shaped dives have a strong negative relationship with distance to the colony; at least until 200 km.

L370-371: This was more evident for U-shaped dives than for late V-shaped dives. The effect that authors highlight here is not that clear to me. I believe that may be some real effect on U-shaped dives, however, for V-shaped dives the effect is visually weak.

DISCUSSION:

After reading the discussion it feels like some studies may have been left behind. I recommend the inclusion of concrete examples in northern gannets or in other gannet species or even boobies. Despite being different species, they have similar foraging and feeding behavior tactics when foraging, thus it can improve substantially the discussion of your results. Moreover, according to the results, it seems quite important the habitat characteristics (e.g., slope) on determining the diving behavior, however, some studies reported that oceanographic conditions can be preponderant as well.

Some possible suggestions:

Cox SL, Miller PI, Embling CB, Scales KL, Bicknell AWJ, Hosegood PJ, Morgan G, Ingram SN, Votier SC. 2016 Seabird diving behaviour reveals the functional significance of shelf-sea fronts as foraging hotspots. R. Soc. open sci. 3: 160317. http://dx.doi.org/10.1098/rsos.160317

Cleasby IR, Wakefield ED, Bodey TW, Davies RD, Patrick SC, Newton J, Votier SC, Bearhop S, Hamer KC (2015) Sexual segregation in a wide-ranging marine predator is a consequence of habitat selection. Mar Ecol Prog Ser 518:1–12. https:// doi. org/ 10. 3354/ meps1 1112

L399: "We found the number of U-shaped dives was greater than expected in 2020." Please avoid this type of sentences in the discussion. This is already stated in the results.

L639-640: Another important variable that might interest the authors might be the distance travelled between two dives. I am just wondering that if adults dive more frequently and used the same diving tactic can be a good indicator of the distribution of prey among and within years. I do not know if your small database (10, 7, and 8 individuals each year) would allow to test this, however it can be very informative. It may help to explain the inter-annual variability in the spatial use of habitat by gannets.

Please check the track changes in the word file. I added some minor comments there and other suggestions that I hope have improved the quality of the manuscript.

Reviewer #3: Thanks to the authors for an interesting and well-written study. The study investigates suitable habitats of forage fishes inferred from the different dive profiles performed by chick-rearing gannets. It identifies important gannet foraging areas and several of their key environmental features using animal-borne biologging data collected over multiple years. The authors propose that a subset of these areas be considered as critical areas of multi-species conservation concern that should be prioritized in protection efforts. In general, the objectives of the study as set out by the authors are achieved and I have no major concerns. Suggestions for improvement are given below.

Please provide higher resolution versions of all figures, especially 3 and 4, these are largely illegible due to blurriness. Consider in figures 3 and 4 to differentiate between land and isobath lines more clearly, for example by using darker grey for land. In addition in figure 3, some panels seem to cut off the southern edge of the distributions.

Line 20: No need to capitalize “Kernel Density”, change to “kernel density”.

Line 33: Replace “incidence on” with “consequences for”.

Line 46: Consider listing one reference per effect (i.e. one for overexploitation by fisheries, one each for climate-induced shifts) to avoid over-referencing.

Line 82: Unclear why “cold-blooded” is relevant here, suggest either deleting or inserting clarification earlier (e.g. at line 77).

Line 90: d’Entremont et al. 2022 a, b or both?

Line 114-115: Assuming devices set to record GPS locations and dive depth. Also please clarify what the range of the base station was.

Line 117: Somewhere at beginning of data processing/analysis please state what software was used (R and R Studio?), and which version.

Line 123-125: Please include a justification (i.e. a reference) for foraging location annotation based on within 30 minute interval between GPS locations (assuming this has to do with transit versus foraging speeds typical for gannets?).

Line 158: No need to redo any analysis, but curious as to why latitude was not included as a covariate.

Line 161-163: Please include information on what the splines were for each covariate (i.e. thin plate regression splines?). Out of interest, were any interaction terms explored for addition to the models, i.e. depth: SST or depth:slope?). Inclusion of interaction terms might help improve deviance explained especially of the V-shaped dive model?

Line 165-170: Please include a reference for the AUC and AIC methods.

Line 188-189: Are the statistical tests of this paragraph described in the methods? If test is among years then surely there should be at least 3 test statistics, i.e. 2019 vs 2020, 2019 vs 2021, 2020 vs 2021? Please also report degrees of freedom.

Line 299: Without going out and testing whether locations of hotspots of U shaped dives contain spawning capelin, or checking corresponding landings by gannets during the early chick-rearing period (i.e. similar to Montevecchi 2007), this is still conjecture, but it is a testable hypothesis for future studies.

Line 343-344: How do you know they are not going for cod during early chick-rearing?

Line 362-364: However, as gannets have been shown to have foraging plasticity, see for example Pettex et al. 2012., and are able to switch over the breeding season i.e. d’Entremont 2022b, it becomes a question of whether the gannets are able to compensate for changing capelin distributions…which is cause for continued monitoring of this colony and situation.

References

Pettex, Emeline, Svein Håkon Lorentsen, David Grémillet, Olivier Gimenez, Robert T. Barrett, Jean Baptiste Pons, Céline Le Bohec, and Francesco Bonadonna. 2012. “Multi-Scale Foraging Variability in Northern Gannet (Morus Bassanus) Fuels Potential Foraging Plasticity.” Marine Biology 159 (12): 2743–56. https://doi.org/10.1007/s00227-012-2035-1.

Reviewer #4: Title: Quantifying space-use of parental Northern Gannets (Morus bassanus) in pursuit of different prey types

This study aimed to assess the distribution and abundance of key forage fishes (namely 1 - capelin, and 2 – other pelagic forage species) in the western North Atlantic by investigating the foraging behaviour and space-use of northern gannets. The study deployed 25 GPS/Time-depth recorder devices on breeding Northern Gannets at Cape St. Mary’s, Newfoundland from 2019 to 2021. Gannets foraging in these waters have been shown to perform two distinctly shaped dives characteristic of the prey they are targeting. U-shaped dives are associated with capture attempts on capelin, while V-shaped dives are linked to other pelagic prey species. The authors used kernel density estimates and habitat suitability models to predict the spatiotemporal distribution of these two different dive types. They created three different habitat suitability models, one for U-shaped dives and two models for V-shaped dives, separated into early and late chick-rearing. The results of the study showed that space-use by gannets varied both within and between years, depending on environmental conditions and the prey they are exploiting. Both dive types generally occurred within shallow coastal waters and cool temperatures (11-15oC), although some V-shaped dives were also performed in waters of ~18oC, coincident with warmer prey species such as Atlantic saury. The authors suggest that regions defined as suitable for U-shaped dives are likely to represent consistent capelin spawning habitat. Given capelin’s importance for many predator species, the authors indicate that these areas, identified as important for spawning capelin, may also be of multi-species conservation concern.

I found this paper to be generally well written, the objectives clearly laid out and the methodology sound. I particularly enjoyed reading the approach for using dive types as a proxy for the distribution of different prey groups. This study makes a nice contribution to the toolkit of approaches for remotely monitoring the state of marine ecosystems. I have made largely minor comments throughout, which I hope the authors will find beneficial.

General comments:

1. This paper considers the distribution of U-shaped and V-shaped dives as a function of environmental conditions and largely ignores the issue of dietary preference. I would like to see some discussion around the effect of dietary preference in determining whether birds would be conducting proportionally more U or V-shaped dives. It seems to me that the study assumes that the preponderance of each dive (prey) type is driven entirely by their availability, without considering a preference for one prey type over another. So, if capelin were the preferred prey, would we not expect the preponderance of U-shaped dives to be nonlinearly related to its availability, remaining the dominant dive type until a threshold after which these dives would be rapidly replaced by V-shaped dives? I think this would be worth some thought, because if there is a level of preference, then clearly the probability of one dive type (V-shaped) occurring is dependent on the probability of the other (U-shaped) not occurring, no? And if there is a confounding effect of preference, this could have implications for how we use such methods to track changes in prey abundance and distribution.

2. The language shifts between present and past tense in a few places. Try to keep tenses consistent throughout.

L. 62. “as asymptomatic population growth”. This comes across as if the colony grew despite years of low productivity. I am not sure if this is what you intended, but either way I think this could be better worded.

L. 70. remove “Clupea harengus” – already defined in L. 55.

L. 71. Change “Parental” to breeding, and remove the later “breeding”

L. 71. Either refer to Newfoundland in full throughout, or define NL with the first reference to Newfoundland.

L. 80. …we aimed…

L. 82. Just refer to prey. To me, the reference to cold-blooded prey implies that they also eat warm-blooded prey, but that you didn’t consider these in this study.

L. 82. …for a better understanding…

L. 102-103. …Uria 300 GPS loggers with Temperature-Depth Recorders…

L. 110. Is the variance metric standard deviation?

L. 112. How do you define a diving bout?

L. 120. It would be helpful to have a figure giving a representative example of a U- and V-shaped dive. This could either be in the main text or in the supplement.

L. 123. This sentence is confusingly worded. Was a dive bout classified as the period of time over which the interval between consecutive dives remained less than 30 min?

L. 126-130. This section could be better explained. It is not altogether clear to me why the authors chose to split V-shaped dives into an early and late season. I think the nature of the influence of U-shaped dives on V-shaped dives could be better described. Would it not be better to account for early/late season effects by fitting the models with a serial autocorrelation structure on date?

L. 140-143. “…we assume that the two dive profiles reflect prey type availability or preference”. I’m not sure I follow this. In my understanding, the distribution of dive shapes would differ considerably depending on whether they represented prey availability or prey preference. For example, if capelin is preferred, you might find the preponderance of U-shaped dives remains dominant even at low capelin densities and high relative availability of other prey. Whereas, if there is no preference, the dominance of dive types would be purely a function of relative availability of their different prey. I think it would be good to at least discuss how the findings here could be influenced by this.

L. 152. This sentence may be better located earlier in the methods. i.e., when describing the fieldwork.

L. 155-160. There should be a reference somewhere to the data products used for these analyses – specifically SST.

L. 159. Were all variables extracted at their native resolution?

L. 160. Were other oceanographic variables considered (e.g. chl a)?

L. 172. It would be good to have reported the mean number of observations (dives) per day, given that predictions were conducted at a daily resolution.

L. 185. …predictions…

L. 188. “association” This could be more clearly stated. It’s not clear whether this implies that they were different or similar.

L. 231-232. Remove “which included distance to the colony, longitude, bathymetry, slope, and sea surface temperature” – this is covered off by the previous sentence.

L. 249-250. Confusingly worded sentence

L 275-276. I’m not sure I agree that an increased number of dives is indicative of decreased prey availability in the case of plunge diving species such as gannets. My understanding is that gannets locate prey from the air and thereafter initiate diving, whereas pursuit divers may undergo multiple exploratory dives to find prey (therefore having to increase dive frequency to locate prey when it is less abundant). Recent work has also found that the number of prey within a prey patch is not a predictor of foraging success (https://doi.org/10.1111/1365-2656.12455) – i.e. they don’t necessarily have to dive more frequently to have success on smaller prey patches. Therefore, I would have expected decreases in prey availability to be associated with concurrent decreases in the frequency of dives.

L. 286. “near” – perhaps towards is a better word choice?

L. 294. “This” – Together, these findings reinforce…

L. 299. “Areas deemed as highly” – Areas deemed highly…

General discussion comments

• There is considerable discussion around the loss of cold-water species (e.g. capelin) from the system and negative effects of this. I would be interested to see some speculation around a possible increased prevalence of warmer water species (e.g. saury), and their potential to (partially) compensate for the loss of colder species. I’m sure they wouldn’t be sufficient to maintain the population, but it might be an interesting discussion point.

• I am wondering if you have access to fisheries data. Given that dive shapes are being linked to different prey types, I think it would be valuable to have the predicted distribution of dives (particularly U-shaped) plotted against the distribution of fishing activity/known spawning grounds. For example, a simple comparison could be to have a kernel density estimate of U-shaped dives overlaid by a kernel density estimate of capelin fishing effort, and/or polygons of their known spawning distribution.

Figures

Fig. 5. It appears that the partial fits for bathymetry show an increase in dive probability over depths of 0 (i.e., over land). Is this just an artefact of inshore foraging close to coastal cliffs? In this case, it would be better to cap all bathymetry values 0 at 0.

6. PLOS authors have the option to publish the peer review history of their article (what does this mean?). If published, this will include your full peer review and any attached files.

Reviewer #1: No

Reviewer #2: No

Reviewer #3: No

Reviewer #4: **Yes: **David Green

---

## [Author Response · Author response to Decision Letter 0]

13 May 2023

A response letter has been uploaded to address all reviewer comments.

---

## [Decision Letter · Decision Letter 1]

21 Jun 2023

PONE-D-23-00624R1Quantifying inter-annual variability on the space-use of parental Northern Gannets (Morus bassanus) in pursuit of different prey typesPLOS ONE

Dear Dr. d'Entremont,

Thank you for submitting your manuscript to PLOS ONE. After careful consideration, we feel that it has merit but does not fully meet PLOS ONE’s publication criteria as it currently stands. Therefore, we invite you to submit a revised version of the manuscript that addresses the points raised during the review process.

We look forward to receiving your revised manuscript.

Kind regards,

Vitor Hugo Rodrigues Paiva, Ph.D.

Academic Editor

PLOS ONE

Journal Requirements:

Reviewers' comments:

Reviewer's Responses to Questions

**Comments to the Author**

1. If the authors have adequately addressed your comments raised in a previous round of review and you feel that this manuscript is now acceptable for publication, you may indicate that here to bypass the “Comments to the Author” section, enter your conflict of interest statement in the “Confidential to Editor” section, and submit your "Accept" recommendation.

Reviewer #2: All comments have been addressed

Reviewer #3: All comments have been addressed

Reviewer #4: All comments have been addressed

2. Is the manuscript technically sound, and do the data support the conclusions?

Reviewer #2: Yes

Reviewer #3: (No Response)

Reviewer #4: Yes

3. Has the statistical analysis been performed appropriately and rigorously? 

Reviewer #2: Yes

Reviewer #3: (No Response)

Reviewer #4: Yes

4. Have the authors made all data underlying the findings in their manuscript fully available?

Reviewer #2: Yes

Reviewer #3: (No Response)

Reviewer #4: Yes

5. Is the manuscript presented in an intelligible fashion and written in standard English?

Reviewer #2: Yes

Reviewer #3: (No Response)

Reviewer #4: Yes

6. Review Comments to the Author

Reviewer #2: I want to congratulate the authors for the thorough revision they carried out. The manuscript is clear, with the statistical procedures and computations clearly stated and understandable. However, on reading through this revised version, I noticed a few typos or very minor grammar issues that I recommend the authors deal with before the acceptance.

I added a few comments and made my correction/suggestions in the file attached.

Reviewer #3: I would like to congratulate the authors on a thorough and well-done revision. My comments have been beyond satisfactorily addressed. Congratulations on this interesting study!

Reviewer #4: I commend the authors on this very interesting study and I appreciate their effort to address the suggestions that I provided on the previous manuscript version. Overall, I am happy with their revisions and feel that they have sufficiently addressed my concerns. I have only a few very minor additional comments (largely typographical) for them to include prior to publication, which I have outline below. Note the Line numbers refer to the tracked changes document.

- L121. …(i.e. from first entry into water until resurfacing)…

- L.146. …early and late V-shaped dives.

- L.369-373. I appreciate that the authors have included this text in response to my previous comment. However, it appears to interrupt the flow and doesn’t really add anything – so maybe better to remove. Perhaps the authors could instead consider changing the wording in L. 365. to: “Increased diving effort in pursuit-diving seabirds, including gannets…”

Also, my understanding is that, in the Angel et al. 2015 paper, the authors only found a significant difference in dive rate for male gannets and only in a single year. So, I’d suggest either changing “diving effort” to “foraging effort”, or removing the Angel et al. 2015 ref.

- L427. …those sites…

- L475. …greater energy on foraging…

- L481. …results in an earlier spring bloom of primary producers…

- L537. …with foraging habitat suitability being heavily reliant on sea surface temperature thresholds of their poikilothermic prey.

- Methods – Habitat Suitability Modelling: I agree with why it wasn’t used, but I think it would still be good to mention why chl a was not included in the analyses – i.e. it was considered, but removed due to poor spatio-temporal coverage over the study period.

7. PLOS authors have the option to publish the peer review history of their article (what does this mean?). If published, this will include your full peer review and any attached files.

Reviewer #2: **Yes: **Ivo dos Santos

Reviewer #3: No

Reviewer #4: **Yes: **David Green

---

## [Author Response · Author response to Decision Letter 1]

27 Jun 2023

Please refer to Reviewer Response Letter attachment.

---

## [Editor Report · Decision Letter 2]

3 Jul 2023

Quantifying inter-annual variability on the space-use of parental Northern Gannets (Morus bassanus) in pursuit of different prey types

PONE-D-23-00624R2

Dear Dr. d'Entremont,

We’re pleased to inform you that your manuscript has been judged scientifically suitable for publication and will be formally accepted for publication once it meets all outstanding technical requirements.

Kind regards,

Vitor Hugo Rodrigues Paiva, Ph.D.

Academic Editor

PLOS ONE
---

## [Editor Report · Acceptance letter]

6 Jul 2023

PONE-D-23-00624R2 

Quantifying inter-annual variability on the space-use of parental Northern Gannets (*IMorus bassanus*) in pursuit of different prey types 

Dear Dr. d'Entremont:

I'm pleased to inform you that your manuscript has been deemed suitable for publication in PLOS ONE. Congratulations! Your manuscript is now with our production department. 

Kind regards, 

on behalf of

Dr. Vitor Hugo Rodrigues Paiva 

Academic Editor

PLOS ONE